# Synergistic Effects of SAP, Limestone Powder and White Cement on the Aesthetic and Mechanical Properties of Fair-Faced Concrete

**DOI:** 10.3390/ma16217058

**Published:** 2023-11-06

**Authors:** Jun Shi, Zhangbao Wu, Jinping Zhuang, Fan Zhang, Tongran Zhu, Huixia Li

**Affiliations:** 1Fujian Communications Planning & Design Institute Co., Ltd., Fuzhou 350000, China; sj936463@163.com (J.S.); 13665039755@163.com (F.Z.); 2Fujian Jiuding Construction Group Co., Ltd., Fuzhou 350000, China; wzb20231013@163.com; 3School of Civil Engineering, Fujian University of Technology, Fuzhou 350118, China; 19781208@fjut.edu.cn; 4Beijing Yihuida Architectural Concrete Construction Engineering Co., Ltd., Beijing 100010, China; ztr18518782068@163.com

**Keywords:** fair-faced concrete, Super Absorbent Polymers (SAP), limestone powder, white cement, aesthetic properties, mechanical properties

## Abstract

In this investigation, a comprehensive assessment was conducted on the cooperative effects of Super Absorbent Polymers (SAP), limestone powder, and white cement within the realm of fair-faced concrete. We discerned that while white cement augments the color vibrancy of the concrete, its accelerated hydration rate potentially induced early-stage cracks and compromised performance. To mitigate these challenges, SAP was incorporated to regulate early hydration, and limestone powder was introduced as a fortifying agent to bolster the mechanical robustness of the concrete. Our findings highlighted not only the capability of SAP to enhance concrete workability and longevity but also the pivotal role of limestone powder in amplifying its mechanical attributes. Microscopic evaluations, undertaken via Scanning Electron Microscopy (SEM), unveiled the potential of both SAP and limestone powder in refining the microstructure of the concrete, thereby elevating its performance metrics. Synthesizing the research outcomes, we pinpointed an optimal amalgamation of SAP, limestone powder, and white cement in fair-faced concrete, offering a valuable reference for prospective architectural applications.

## 1. Introduction

In recent years, fair-faced concrete has garnered widespread attention in the architectural realm due to its natural, minimalist appearance and superior engineering properties [1]. With advancements in the design of public structures and high-end residences, the demand for high-quality and visually appealing fair-faced concrete has been on the rise [2,3]. White cement, in particular, owing to its exceptional color stability and high reflectivity, has emerged as the material of choice for producing such concrete [4,5,6,7,8]. However, achieving the desired outcome with fair-faced concrete is not solely reliant on the quality of cement [9,10,11,12]; the synergistic effects of other components within the concrete are equally pivotal.

Super Absorbent Polymers (SAP) have taken center stage in concrete research [13,14]. Their prominence is not just because they can serve as internal curing agents, providing continuous moisture supply [15,16], ensuring more uniform hardening, but also because they can significantly reduce the drying shrinkage of concrete, thereby enhancing its durability [17,18,19]. Furthermore, SAP play a key role in modulating the hydration process of concrete [20]. They possess the capability to absorb vast amounts of water and release it as the concrete starts drying, helping in the mitigation of crack formation [21,22,23,24]. The inclusion of SAP has been noted to considerably improve both the compressive and flexural strength of concrete, making it more adaptable for various architectural applications [25,26,27,28]. Moreover, SAP interact synergistically with other constituents in concrete, like aggregates and admixtures, further refining its performance [29,30].

Similarly, the role of limestone powder in concrete is undeniable [31,32]. It not only acts as a mineral admixture, enhancing the workability and eventual mechanical properties of concrete [33,34], but its unique chemical properties also influence the curing process of concrete [35,36]. The inclusion of limestone powder has been proven to augment the impermeability and durability of concrete [37], which is crucial for structural elements that remain exposed to harsh conditions for extended durations. Moreover, the combined use of limestone powder and SAP can produce synergistic effects, further elevating the performance of concrete [38].

Nevertheless, despite the positive impact of both SAP and limestone powder on the performance of fair-faced concrete, questions remain regarding their interaction with white cement and how, collectively, they influence both the aesthetics and mechanical properties of the concrete [39]. Recent studies have initiated explorations into the interactions of these materials and their effects on concrete performance, yet a comprehensive theoretical framework to guide practice remains elusive [32,40].

In conclusion, the use of white cement in fair-faced concrete has been deeply researched and widely applied. Simultaneously, both SAP and limestone powder are receiving increasing attention due to their beneficial effects on concrete performance. However, the synergistic utilization of these three materials to achieve optimal fair-faced concrete performance remains a challenge. This study aims to delve deep into the interactions of these materials within fair-faced concrete, hoping to offer more insights and recommendations for practice.

## 2. Materials and Methods

### 2.1. Materials

#### 2.1.1. Fundamental Properties of the Material

In this study, white Portland cement (strength grade 42.5 and fineness of 300 mesh) was sourced from Aalborg Portland (Anqing) Co., Ltd., Anqing, China. The research also utilized Class II fly ash and Ordinary Portland cement, the chemical composition of which is detailed in Table 1. The fine aggregate employed was medium sand, characterized by a fineness modulus of 2.7. The coarse aggregate comprised graded gravel particles, specifically within the size ranges of 5–10 mm and 10–25 mm. The water-reducing agent incorporated into the mix was a polycarboxylate superplasticizer generously provided by Xiamen Lets Group. Super absorbent resin with a particle size ranging from 30–60 mesh was procured from Yixing’s Kexin Chemical Co., Ltd., Yixing, China. The main component of SAP is low cross-linked polyacrylic acid sodium salt, and the main performance indexes are shown in Table 2. Additionally, limestone powder with a fineness of 300 mesh was incorporated, sourced from the Tianjin Guangfu Fine Chemical Research Institute.

#### 2.1.2. Mixture Design

The control group [41,42,43,44], denoted as W-0, was selected, and the white cement content was adjusted based on W-0 to prepare white fair-faced concrete. W stands for white cement mixed with fair-faced concrete specimen. Additionally, 25, 50, and 100 represent the white cement content. SW stands for SAP mixed with white fair-faced concrete. An optimal dosage of SAP exists. Lower dosages fail to achieve the material’s internal curing effect, while higher dosages may impair the concrete’s mechanical properties due to changes in the actual water–cement ratio and increased porosity. Therefore, SAP content was set at 0.2%, 0.3%, 0.4%, and 0.5% of the total mass of the cementitious material, according to the literature [45,46,47]. The optimal dosage of pre-absorbed water, as described in the literature [39,48,49], was determined to be 10 times the mass of SAP. Therefore, pre-absorbed water was adjusted to be 5, 10, and 15 times the mass of SAP in order to prepare white fair-faced concrete. LSW stands for limestone powder SAP mixed with white fair-faced concrete specimen. Additionally, 10, 20, and 30 represent the limestone powder content, which replaces the quality of the cementitious material, respectively, 10%, 20%, and 30%. The mix ratios of the specimens are summarized in Table 3.

### 2.2. Experimental Methods

#### 2.2.1. Sample Preparation

During the sample preparation, white cement, water, coarse aggregate, and fine aggregate were initially combined to ensure a homogenous mixture. During this mixing phase, Super Absorbent Polymer (SAP) was incrementally added based on the desired proportion, ensuring its even dispersion throughout the concrete slurry. Limestone powder was subsequently incorporated in a stepwise manner, with continued mixing to ascertain its thorough integration with the slurry. Once the mixture was prepared, it was transferred to pre-arranged molds. To achieve an even distribution within the mold and to eliminate potential air bubbles, a vibrator was employed. The poured concrete was then subjected to a controlled curing environment with consistent temperature and humidity for a duration of 28 days, ensuring the concrete fully solidified and reached its optimal performance characteristics. Post-curing, the concrete specimens were demolded from their molds.

#### 2.2.2. Slump Test

The workability of fair-faced concrete was investigated using the slump test. This test was rigorously conducted following the protocols set out in the “Standards for Performance Tests of Ordinary Concrete Mixtures (GB/T50080-2016)” [50]. During the testing process, the slump cone was first securely positioned on a flat base. The concrete sample was then introduced into the cone in three stages: 1/3, 2/3, and full cone. After each stage, 25 strokes of rod tamping were consistently applied. Upon completion, the slump cone was carefully removed, and the diameter of the slumped concrete was measured. It is noteworthy that the slump value, determined by the difference between the maximum diameter of the slumped concrete and the base diameter of the cone, is indicative of the concrete’s workability; a larger slump value signifies superior workability.

#### 2.2.3. Strength Tests

The standard cube compressive strength and splitting tensile strength tests were carried out in accordance with the Standards for Test Methods of Mechanical Properties of Ordinary Concrete (GB/T50081-2002) [51]. Cubic specimens (150 × 150 × 150 mm) of fair-faced concrete mixed with white cement and SAP were prepared, with three specimens in each group. The specimens were cured for 3, 7, and 28 days in a constant temperature and humidity curing room, after which their cubic compressive strength and splitting tensile strength were measured. The maximum load was recorded and used to calculate the concrete’s compressive strength and splitting tensile strength.

#### 2.2.4. Rapid Chloride Permeability Test

This study performed the Rapid Chloride Permeability Test (RCPT) following the Chinese Standard for Long-term Performance and Durability of Ordinary Concrete (GBT50082-2009) [52]. Cylindrical specimens with a diameter of (100 ± 1) mm and a height of (50 ± 2) mm were prepared, with three specimens in each group. They were cured for 28 days. The RCPT test measures the electrical conductivity of concrete by recording the RCPT value (C) through the test over a period of 6 h at 60 V, which is used to calculate the converted RCPT value for each specimen.

#### 2.2.5. Scanning Electron Microscopy Test

This study conducted SEM tests to observe the microstructure characteristics of fair-faced concrete at different magnifications. Samples with a length of 1 cm, a width of 1 cm, and a thickness of 3 mm were prepared from the core part of specimens cured for 28 d. One side was smoothed with sandpaper to serve as the bottom for sticking, and the other side was used for viewing under SEM (TESCAN MIRA LMS).

The specific experimental scheme and procedure are illustrated in Figure 1 below.

## 3. Results and Discussion

### 3.1. Influence of SAP, Limestone, and White Cement on Fair-Faced Concrete

#### 3.1.1. Effect on Workability

(1)The impact of SAP on the workability of fair-faced concrete

From Figure 2a, it is evident that in a study on the workability of white cement fair-faced concrete, the incorporation of SAP played a crucial role in dictating its behavior. In particular, when the SAP was introduced at a concentration of 0.0%, the slump of the concrete was gauged at 170 mm. Intriguingly, as the SAP concentration escalated to 0.4%, the slump witnessed a rise, reaching 210 mm, signifying enhanced workability. Further, SEM (Scanning Electron Microscopy) analysis showcased that the addition of SAP led to a more homogeneous microstructure in the concrete. Relative to the concrete devoid of SAP, a pronounced decrease in internal voids was observed. Such outcomes highlight the potential of SAP in refining the workability and microstructure of white cement fair-faced concrete. However, the precise concentration of SAP is imperative to harness its benefits effectively.

(2)The impact of water absorption on the workability of fair-faced concrete

From Figure 2b, in the conducted experiments, it was observed that with SAP (Super Absorbent Polymer) water absorption ratio of 0, the slump of the concrete was 170 mm. However, with an increase in the water absorption ratio of SAP to 5, the slump slightly increased to 175 mm. Intriguingly, as the water absorption ratio further escalated to 10, the slump value experienced a noticeable rise, reaching 185 mm. An even more pronounced effect was seen when the ratio was adjusted to 15, where the slump peaked at 200 mm, showcasing enhanced workability. These findings clearly indicate that the water absorption capacity of SAP can significantly influence the workability of concrete. Adjusting the water absorption ratio of SAP can be a viable strategy to optimize the workability properties of concrete, emphasizing the importance of carefully controlling SAP’s water absorption characteristics to achieve the desired effects.

(3)The impact of limestone on the workability of fair-faced concrete

From Figure 2c, in the conducted experiments, it was observed that without the addition of SAP and without any limestone powder, the slump of the concrete was 170 mm. When 10% limestone powder was added under the same SAP dosage, the slump increased to 206 mm. This indicates that the combined effect of limestone powder and SAP further improved the workability of the concrete. However, with an increase in the SAP addition to 0.4% and without any limestone powder, the slump was measured to be 210 mm. These data clearly demonstrate that the appropriate addition of SAP and limestone powder can significantly enhance the workability of white cement fair-faced concrete. However, the precise amount and ratio need careful adjustment to ensure optimal effects.

#### 3.1.2. Impact on Durability

The criteria taken from the Chinese Standard for inspection and assessment of concrete durability (JGJ/T193-2009) [53] are presented in Table 4, which provides a clear categorization of permeability based on the RCPT values. From the results shown in Figure 3, it can be observed that the RCPT values of the plain concrete with added admixtures are all below 100 C, meeting the requirements of Level 4. Especially after the addition of limestone, it is less than 500 C, achieving the requirements of Level 5. This observation underscores the efficacy of the admixtures in enhancing the properties of the concrete.

(1)Impact of white cement on the durability of fair-faced concrete

Figure 3a shows that with the increase of white cement content, the electrical flux value first decreases and then increases. Experimental data indicates that when the white cement content is 50%, the electrical flux value is the smallest, while the electrical flux value is the largest when it is pure white cement. These results suggest that the optimal durability of fair-faced concrete is achieved when the white cement content is around 50%. The reason for this is that the addition of 0–50% white cement accelerates the early hydration of fair-faced concrete, resulting in faster hydration and hardening, an increase in density, and a decrease in RCPT value. However, as the white cement content increases, the cement’s hydration rate accelerates, the heat of hydration increases, and voids form inside the concrete, leading to an increase in RCPT value and a reduction in the stability of fair-faced concrete. Therefore, to meet the durability requirements, the optimum white cement dosage is 50%.

(2)Impact of SAP on the durability of fair-faced concrete

Figure 3b shows that as the SAP content increases, the RCPT value of white cement clear water concrete first decreases and then increases. When the SAP content is 0.3%, the RCPT value is 498.15 C, which is 31.9% lower than the 731.95 C during dry mixing. However, when the content is 0.5%, the RCPT value is 677.01 C, an increase of 35.9%. The smallest RCPT value is observed at a dosage of 0.3%. As white cement accounts for 50% of the total cement in the test material, it hydrates rapidly. Adding an appropriate amount of SAP can delay cement hydration, but neither an excessive nor an insufficient amount of SAP can achieve this goal.

The inconsistencies observed in these results could stem from several factors. Firstly, when the SAP content surpasses the optimal threshold, the SAP particles might become oversaturated, leading to the creation of minute voids. These minuscule cavities could serve as pathways of minimal resistance, resulting in a heightened RCPT value due to increased permeability. Secondly, while an adequate amount of SAP can offer internal curing to the concrete, thus delaying cement hydration, both an excessive and insufficient quantity of SAP may disrupt the concrete’s microstructure, leading to compromised physical properties. These elements collectively account for the inconsistent RCPT values observed at certain SAP dosages.

Figure 3c shows that as the SAP water absorption ratio increases, the RCPT value gradually decreases. When the SAP water absorption ratio is 15 times, the electrical flux value has been reduced to 457.87 C, which is 37.4% lower than the 731.95 C of dry-mixed white cement clear water concrete. Insufficient water leads to high viscosity of the cement paste, making it difficult to stir evenly. This is consistent with the description of Cai [1]. As a result, numerous voids exist in the concrete after hardening, and the density of the hydrated colloid structure is low, leading to a high RCPT value when five times water absorption rate. As the water absorption rate increases, the cement hydrates fully, gels and crystals form, and the density increases, resulting in a decrease in the RCPT value. As shown in Figure 3c, the RCPT value continues to decrease with an increase in the water absorption rate, indicating that a water absorption rate of 15 does not exceed the optimal value.

(3)Impact of limestone on the durability of fair-faced concrete

Figure 3d showcases the influence of limestone powder content on the electric flux of SAP-enriched white fair-faced concrete. Relative to the control, the electric fluxes for LSW-10, LSW-20, and LSW-30 manifest reductions of 7.4%, 10.2%, and 17.6%, respectively. These findings underscore the efficacy of limestone powder in augmenting the durability of SAP-integrated white fair-faced concrete. Intriguingly, an increase in limestone powder content correlates with enhanced durability of the fair-faced concrete. The underlying mechanism can be attributed to the micro-aggregate filling effect of limestone powder. Its granulometry is finer than both cement and fly ash, thereby conferring a denser slurry and refining the internal matrix of the concrete. Furthermore, limestone powder exhibits a nucleation effect, serving as an initiation platform for C-S-H crystallization. This reduces the energy barrier for nucleation, promoting the prolific formation of C-S-H crystals. Thus, the incorporation of limestone powder into SAP-based fair-faced concrete offers a promising avenue to bolster its durability.

#### 3.1.3. Impact on Mechanical Properties

(1)Influence of white cement on the strength of fair-faced concrete

Fair-faced concrete samples were prepared by mixing ordinary cement with white cement at mass ratios of 0%, 25%, 50%, 75%, and 100%. Compressive strength tests and split tensile strength tests were performed on the samples after curing for 3 d, 7 d, and 28 d. The test results are shown in Figure 4a and Figure 4b, respectively.

Figure 4a depicts the compressive strength of fair-faced concrete at different curing periods. The results show that the 3-day compressive strength of fair-faced concrete exceeds 41 MPa, indicating rapid strength gain during the early curing period. Furthermore, the 3-day and 7-day compressive strength increases as the proportion of white cement in the mix increases. However, at a curing period of 28 days, the compressive strength of fair-faced concrete tends to decrease as the white cement content increases. Compared to ordinary cement, fair-faced concrete containing white cement exhibits higher early strength but lower later-stage strength. These observations can be attributed to the higher C_3_S content in white cement, which accelerates the hydration reaction rate and promotes early strength gain in fair-faced concrete [54]. The early strength of fair-faced concrete increases with increasing white cement content, but the rate of increase slows down as the curing time lengthens. Consequently, the later-stage strength of fair-faced concrete is lower than that of ordinary fair-faced concrete.

Figure 4b presents the split tensile strength of fair-faced concrete at different curing periods. The results indicate that white cement fair-faced concrete exhibits rapid early strength gain, with the 3-day tensile strength reaching 2.35 MPa. Moreover, the 3-day tensile strength of white cement fair-faced concrete remains almost constant even with an increase in the proportion of white cement, whereas the tensile strength of ordinary cement fair-faced concrete increases by 1.3% to 2.46 MPa. This observation suggests that white cement has a faster hydration reaction rate, which promotes earlier strength gain. Additionally, the increase in tensile strength during the 3-day to 7-day curing period is less than that during the 7-day to 28-day period, similar to the pattern observed for the compressive strength of fair-faced concrete. By the age of 28 days, the tensile strength of white fair-faced concrete exceeds that of ordinary cement fair-faced concrete, indicating excellent mechanical properties. These results suggest that white cement can be used to alter the color of fair-faced concrete without compromising its mechanical properties.

(2)Effect of SAP content on the mechanical properties of fair-faced concrete

To improve the quality of fair-faced concrete made with a combination of white cement and ordinary cement, a study was conducted where varying amounts of SAP were added. The cement content used was 50%, and SAP in concentrations of 0%, 0.2%, 0.3%, 0.4%, and 0.5% were included after being pre-absorbed ten times. The effect of SAP concentration on the strength of the resulting fair-faced concrete at different ages is presented in Figure 5a,b.

Figure 5a shows that the addition of pre-absorbed SAP results in a varying degree of reduction in compressive strength of white fair-faced concrete, with the compressive strength accounting for 93.6% and 63.9% of the benchmark group. This finding is consistent with the literature [55], which reports that the addition of SAP to concrete reduces its strength at different ages, and the degree of strength reduction increases with the dosage of SAP. Additionally, the reduction in compressive strength is more significant at earlier ages. However, the degree of strength reduction at 28 d is much smaller than that at 3 d and 7 d. The literature [55] also suggests that the addition of pre-absorbed water SAP delays the early (3 d) hydration reaction of cement while promoting its hydration in the middle and late stages (7 d, 28 d), given that the effective water–cement ratio remains the same. Moreover, the addition of SAP increases the total water–cement ratio of concrete, which causes the water-absorbed SAP particles to lose water and collapse, forming larger pores that reduce the strength of the concrete [55,56]. Additionally, SAP delays the early cement hydration [17,28], resulting in a significant decrease in the early strength of concrete after SAP is added. However, as hydration progresses, the water in SAP is gradually released for cement hydration reaction, increasing the degree of cement hydration, and consequently leading to a significant increase in the later strength of concrete.

Figure 5b indicates that the tensile strength at 3 d and 7 d consistently decreases as the SAP content increases, remaining lower than the reference value. This is because the pre-absorbed SAP passively increases the water–cement ratio during the early stage, leading to increased porosity and decreased tensile strength while delaying early cement hydration.

(3)Effect of SAP water absorption ratio on fair-faced concrete

The pre-absorbed water ratio refers to the ratio of the weight of SAP after pre-absorption to the weight of SAP. In this study, fair-faced concrete with SAP content of 0.2% was created by adding SAP with pre-absorbed water ratios of 5, 10, and 15 to a cement mixture containing 50% white cement and 50% ordinary cement. The findings are illustrated in Figure 6a,b.

Figure 6a shows that the compressive strength of concrete decreases as the water absorption ratio increases. This indicates that excess water cannot be absorbed by SAP, leading to an increase in the effective water–cement ratio and a decrease in the compressive strength of concrete. Hence, determining the appropriate water absorption ratio is crucial for improving compressive strength.

Figure 6b illustrates that the tensile strength of fair-faced concrete is initially low due to the delayed hydration reaction rate of SAP. However, it rapidly increases during the middle and late stages of curing (7 d, 28 d), which is consistent with the compressive strength performance discussed earlier.

(4)Effect of limestone powder on the mechanical properties of fair-faced concrete

As depicted in Figure 7a, with a limestone powder content of 20%, the compressive strengths of white fair-faced concrete mixed with SAP at curing ages of 3 days, 7 days, and 28 days are enhanced by 9.8%, 16.7%, and 13.6% respectively. These findings align with the research outcomes presented by Li [57], Huo [58], and Ghafoori [59]. The underlying rationale is the filler effect of the limestone powder, which refines the pore structure of the concrete. This refinement counteracts the potential increase in porosity instigated by the integration of SAP, thereby promoting a denser internal structure and amplifying the compressive strength.

However, at a limestone powder content of 30%, there is a slight reduction in the compressive strength of the SAP white fair-faced concrete at the 3-day and 7-day marks. This can be attributed to the intensification of the dilution effect as the limestone powder content rises. Consequently, the early-stage hydration products of the white fair-faced concrete surpass those of the standard white fair-faced concrete. Nevertheless, as the curing age progresses, a notable increase of 11.6% in compressive strength is observed at 28 days. Based on the 28-day standard curing results, the inclusion of limestone powder is evidently advantageous for enhancing the compressive strength of SAP white fair-faced concrete. A limestone powder content of 20% yields the most prominent overall strength enhancement.

Figure 7b elucidates the impact of limestone powder content on SAP-mixed white fair-faced concrete. With a limestone powder content of 30%, the tensile strengths at various intervals exhibit increases of 17.6%, 10.5%, and 12.1%. Clearly, both the filler and nucleation effects of limestone powder contribute positively to the tensile strength of the SAP-mixed white fair-faced concrete [34].

#### 3.1.4. ANOVA Analysis of the Influence of Various Factors

To further delve into the statistical significance of the effects of different factors such as SAP concentration, limestone content, and white cement content on the properties of fair-faced concrete, an Analysis of Variance (ANOVA) was conducted [60].

The factors were treated as independent variables, while the properties of the concrete, including slump value, RCPT value, compressive strength, and tensile strength, were treated as dependent variables. The main objective was to determine if the variations in the means of these properties were significant across different levels of the factors. In statistics, the *p*-value is a measure that helps determine the significance of the results. A smaller *p*-value indicates stronger evidence against a null hypothesis, suggesting the factor has a significant effect. Typically, a *p*-value less than 0.05 is considered statistically significant. The *p*-value calculation results are shown in Table 5.

It can be seen from Table 5 that the *p*-value between the SAP concentration and each response variable was below the conventional significance level (0.05), indicating that the SAP concentration has a statistically significant effect on the workability, durability, and mechanical properties of the concrete. The ANOVA results showed that the limestone content also had a significant influence on the properties of the concrete, especially on its compressive and tensile strengths. The variance analysis indicated that white cement content significantly affected the durability of the fair-faced concrete, especially its RCPT value.

### 3.2. Effects on Color and Appearance of Fair-Faced Concrete

#### 3.2.1. Impact of White Cement on the Color of Concrete

Through a series of experimental studies, it can be seen from Figure 8 that white cement significantly influences the color of fair-faced concrete. The unique color attributes of white cement enhance the concrete’s hue noticeably when it partially or entirely substitutes ordinary Portland cement, catering to the aesthetic demands of white fair-faced concrete. Additionally, the greater amount of C_3_A produced during the hydration process of white cement might be associated with its distinct color and an increase in early strength. However, it is worth noting that while white cement can amplify the color saturation of concrete, its faster hydration rate might adversely affect other properties of the concrete, potentially leading to early cracking. Despite these challenges, due to its superior color and aesthetic appeal, the application of white cement in architectural decoration, road and bridge construction, and other aesthetically demanding areas continues to grow.

#### 3.2.2. Impact of SAP and Limestone Powder on the Appearance of Concrete

It can be seen from Figure 9 that the application of SAP and limestone powder in concrete significantly impacts its appearance. Firstly, the inclusion of SAP facilitates internal curing of the concrete, enhancing its visual appeal and surface texture. This improvement is primarily attributed to the high water-absorbing capacity of SAP, providing a continuous water source to prevent drying cracks in the concrete. Secondly, limestone powder, as an additive, not only ameliorates the workability of the concrete but also elevates its visual consistency and color uniformity. This improvement arises from the limestone powder’s ability to enhance the interfacial bonding between cement and aggregate at the microscopic level, rendering a more uniform and smoother surface for the concrete. In summary, the combined application of SAP and limestone powder showcases marked advantages in elevating the aesthetic quality and appearance of concrete [61,62].

#### 3.2.3. Color Difference Quantitative Analysis

Two types of concrete samples were selected for comparison in the study. Sample A utilized traditional ordinary Portland cement, while Sample B incorporated white cement. To ensure the accuracy of the test, all other ingredients and curing conditions for both samples were kept consistent. A high-precision colorimeter, capable of measuring L, a, and b values, was employed to determine the color of the concrete samples. Initial measurements of Sample A yielded L_1_ = 60, a_1_ = 0.5, and b_1_ = 1.5, while subsequent measurements for Sample B produced L_2_ = 80, a_2_ = 0.2, and b_2_ = 1.0. The color difference between the two samples was then calculated using the respective color difference formula.
$$\Delta E=\sqrt{{\left(L_{2}-L_{1}\right)}^{2}+{\left(a_{2}-a_{1}\right)}^{2}+{\left(b_{2}-b_{1}\right)}^{2}}$$

After inputting the values, we obtain:$$\Delta E=\sqrt{{\left(80-60\right)}^{2}+{\left(0.2-0.5\right)}^{2}+{\left(1.0-1.5\right)}^{2}}=20.1$$

The color difference value ΔE = 20.1 indicates a significant color variation between the two concrete samples. This is primarily attributed to the use of white cement, which results in Sample B having a whiter and brighter color than Sample A. Given that the color difference surpasses the commonly accepted range (typically between 2.0 and 3.0), it can be concluded that white cement has a pronounced effect on the color of fair-faced concrete.

### 3.3. Microstructural Analysis of Fair-Faced Concrete Using SEM

#### 3.3.1. Results of SEM

Scanning Electron Microscopy (SEM) was employed to investigate the microstructural changes in fair-faced concrete with and without the incorporation of Super Absorbent Polymer (SAP). All SEM micrographs were acquired under consistent imaging conditions to ensure comparability. Details of the accelerating voltage and magnification are provided in the respective figure captions.

Figure 10a displays the inherent microstructure of fair-faced concrete in the absence of SAP, characterized by a relatively uniform matrix. In contrast, Figure 10b highlights the pronounced microstructural alterations induced by SAP. Specifically, circular pores, annotated as ‘A’, and larger macropores with diameters up to several hundred microns, labeled as ‘B’, are prominent. In comparison to the SW group in Figure 10b, the LSW group in Figure 10c exhibits a marked reduction in surface pore density. The incorporation of limestone powder serves as an effective countermeasure to the pore enlargement typically induced by the addition of SAP. This modulatory effect of limestone powder mitigates the adverse influence of SAP on the mechanical characteristics of white fair-faced concrete, culminating in enhancements in both compressive and tensile strengths.

A closer examination of the regions surrounding the spherical fly ash particles is facilitated by Figure 10d,e. Figure 10d reveals a scattered presence of hydration products in the SAP-free concrete, failing to culminate in a dense structure. In stark contrast, Figure 10e illustrates the gradual release of water from SAP, catalyzing the cement hydration process and leading to a densely aggregated hydration product formation.

In Figure 10f, images of the SAP-composite white cement plain concrete incorporated with limestone powder, there is a noticeable increase in the formation of Calcium–Silicate-Hydrate (C-S-H) and Calcium Hydroxide (Ca(OH)_2_). The enhanced presence of these hydration products suggests that the limestone powder, in synergy with the SAP, has promoted the hydration process of the white cement. This microstructural observation underscores the potential benefits of introducing limestone powder into SAP-composite white cement concrete, as it appears to augment the density and integrity of the matrix, potentially translating to improved mechanical properties and durability.

#### 3.3.2. Microscopic Interpretation of Macroscopic Performance

(1)Microstructure and mechanical properties

The strength and durability of concrete largely depend on its pore structure. Smaller, evenly distributed pores help enhance the compactness and mechanical properties of concrete. The addition of SAP can influence the pore structure. In the early stages of curing, the SAP releasing water might increase porosity, but in the later stages, due to the continuous hydration water provided by SAP, it helps fill these pores, thereby improving strength.

White cement and limestone powder react with water in the presence of moisture, forming hydrated calcium silicate (C-S-H) and other hydration products. These hydration products fill the voids in the original mixture, thereby strengthening the structure of the concrete. Specifically, the inclusion of SAP provides additional moisture, facilitating the formation of more C-S-H gel, which in turn enhances the strength and durability of the concrete.

(2)Microstructure and durability

Electrical flux is commonly used to evaluate the compactness and durability of concrete. A lower electrical flux indicates better compactness and durability. From the documentation, the combined use of white cement and SAP can significantly reduce electrical flux, which microscopically indicates a better pore structure and distribution of hydration products.

The durability of concrete is also related to the size and distribution of pores. Smaller pores and even distribution help prevent the penetration of corrosive substances, thereby enhancing the durability of the concrete. The addition of SAP and limestone powder helps optimize the pore structure, thereby strengthening the concrete’s durability.

(3)Microstructure and impermeability

The impermeability of concrete is related to the connectivity of the pores. The lower the pore connectivity, the better the impermeability of the concrete. The addition of SAP and limestone powder helps reduce pore connectivity, thereby enhancing the concrete’s resistance to permeation.

The macroscopic properties of fair-faced concrete, such as strength, durability, and impermeability, are influenced by its microstructure. The pore structure, formation and distribution of hydration products, and pore connectivity all affect the performance of concrete at the microscopic level. The addition of SAP and limestone powder helps optimize these microstructures, thereby enhancing the macroscopic properties of fair-faced concrete.

#### 3.3.3. Relationship between Microstructure and Aesthetic Properties

(1)Microstructure of SAP and Fair-Faced Concrete

The primary function of SAP is to absorb and store a significant amount of moisture within the concrete and gradually release this moisture during the curing process. This process impacts the microstructure of the concrete, particularly the distribution and size of the pores. In the early stages of concrete curing, the water released from SAP might lead to an increase in porosity, affecting the compactness of the concrete at the microscopic level. This could influence the concrete’s appearance, resulting in more visible pores or cracks on its surface. However, in the later stages of curing, the moisture provided by SAP supports the continuous hydration of the cement, filling these pores, thereby enhancing the concrete’s compactness and visual uniformity.

(2)Microstructure of Fair-faced Concrete with Limestone Powder

Limestone powder, as a mineral admixture, can enhance the microstructure of concrete. Its filler effect aids in reducing porosity and increasing the compactness of the concrete. Aesthetically, limestone powder can render a smoother and more uniform surface to the concrete. Additionally, it can impart a more consistent coloration, contributing to a more visually appealing appearance of the concrete.

(3)Relationship between Microstructure and Aesthetic Properties

The microstructure of concrete, especially the distribution and size of its pores, directly influences its appearance. Concrete with fewer pores and uniform distribution typically presents a superior appearance devoid of extensive cracks and voids. The incorporation of SAP and limestone powder both impact the microstructure of the concrete, consequently affecting its visual characteristics. Specifically, they both can enhance the compactness of the concrete, leading to improved surface uniformity and aesthetic appeal.

The aesthetic properties of fair-faced concrete are intricately linked to its microstructure. Both SAP and limestone powder, as admixtures, can enhance the appearance of concrete by improving its microstructural attributes. Specifically, they both contribute to increasing the compactness of the concrete, resulting in a more uniform and visually appealing appearance.

#### 3.3.4. Relationship between Microstructure and Mechanical Properties

(1)Influence of SAP on the Microstructure of Fair-faced Concrete

The primary role of Super Absorbent Polymers (SAP) in concrete is to absorb substantial amounts of water and gradually release this moisture during the concrete’s curing process. This released moisture creates minute pores, thereby affecting the concrete’s structure at the microscopic level. After the addition of SAP, the 28-day electrical flux of white cement fair-faced concrete is lower than that of concrete without SAP. This suggests that the microstructure of the concrete with SAP is more compact with fewer pores.

(2)Effect of Limestone Powder on the Microstructure of Fair-faced Concrete

Limestone powder, as a mineral admixture, serves to fill minute voids in concrete, thereby enhancing its microstructural compactness. This filling effect contributes to the overall mechanical performance of the concrete. When the limestone powder content is at 30%, combined with 0.3% SAP, the concrete’s mechanical properties are optimized. This underscores the significance of limestone powder in refining the microstructure of concrete.

(3)Relationship between Microstructure and Mechanical Properties

The mechanical properties of concrete, such as compressive strength and tensile strength, are largely dependent on its microstructure. Concrete with smaller, uniformly distributed pores typically exhibits higher strength. The addition of both SAP and limestone powder impacts the microstructure of the concrete. Specifically, they both contribute to enhancing the compactness of the concrete, subsequently improving its mechanical performance. The microstructure of fair-faced concrete is intricately linked to its mechanical properties. Both SAP and limestone powder can enhance the mechanical performance of concrete by optimizing its microstructure. For instance, the data mentioned above indicates that when the SAP content is at 0.3% and the limestone powder content is at 30%, the 28-day compressive strength of the concrete is about 20% higher than other combinations. This significant enhancement can be attributed to the ameliorating effects of SAP and limestone powder on the microstructural attributes of the concrete.

## 4. Conclusions

(1)The Superabsorbent Polymer (SAP) is pivotal in white cement fair-faced concrete. This is largely attributed to its ability to regulate the hydration of the concrete, especially in the early stages. By gradually releasing moisture during the curing process, SAP ensures continuous hydration of the cement. This is crucial as white cement, while enhancing the color vibrancy of concrete, has an accelerated hydration rate that could lead to early-stage cracks. By mitigating this challenge, SAP ensures the longevity and workability of the concrete. Through extensive testing and evaluations, SAP content of 0.3% has emerged as the optimal proportion for delivering superior concrete performance.(2)Limestone powder serves a dual purpose in the realm of fair-faced concrete. Firstly, as a mineral admixture, it significantly bolsters the mechanical properties of the Concrete, acting as a fortifying agent. Secondly, its presence refines the concrete’s microstructure, as evidenced by Scanning Electron Microscopy (SEM) evaluations. These microscopic evaluations have unveiled the profound impact of limestone powder on the concrete’s performance metrics. Through a series of experiments, a combination of 30% limestone powder content with 0.3% SAP has been determined to provide the optimal comprehensive performance for the concrete.(3)The beauty of fair-faced concrete is not just skin-deep. Its aesthetic properties are intricately linked to its microstructure. The uniformity, color vibrancy, and overall aesthetic appeal of the concrete are significantly dictated by the fine-tuning of its microstructure. By optimizing the proportions of SAP and limestone powder, not only is the mechanical robustness of the concrete enhanced, but its aesthetic appeal also sees marked improvement.(4)The results of the ANOVA reinforce the observations made in the previous sections. The significant *p*-values associated with the factors underscore their influence on the properties of fair-faced concrete. For instance, the significance of SAP concentration aligns with its observed effect on workability and microstructure. Similarly, the importance of limestone content in enhancing mechanical strengths is statistically validated.(5)This investigation is more than just a theoretical exploration; it holds pragmatic value. By shedding light on the synergistic effects of SAP, limestone powder, and white cement, the research offers invaluable insights into the optimal formulations for white cement fair-faced concrete. These findings are not just academically significant but hold substantial relevance for real-world engineering applications. Architects, civil engineers, and construction professionals can leverage these insights to produce superior quality fair-faced concrete, combining aesthetic appeal with mechanical robustness.

## Figures and Tables

**Figure 1 materials-16-07058-f001:**
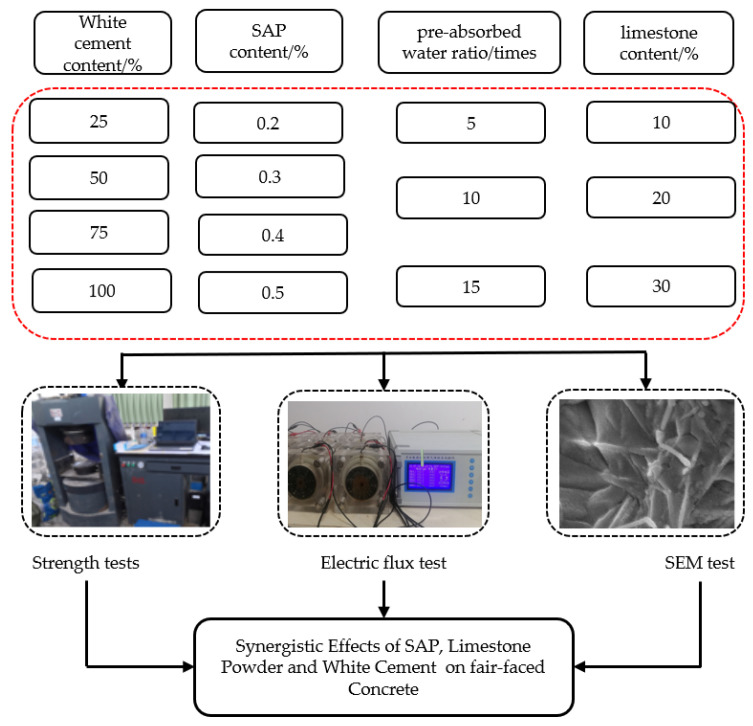
Flowchart of experimental scheme and procedure.

**Figure 2 materials-16-07058-f002:**
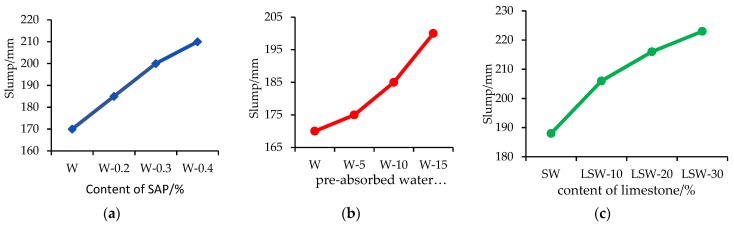
Influence of Different Factors on Slump. (**a**) Slump variation with SAP content (**b**) Slump variation pre-absorbed water ratio. (**c**) Slump variation with content of limestone.

**Figure 3 materials-16-07058-f003:**
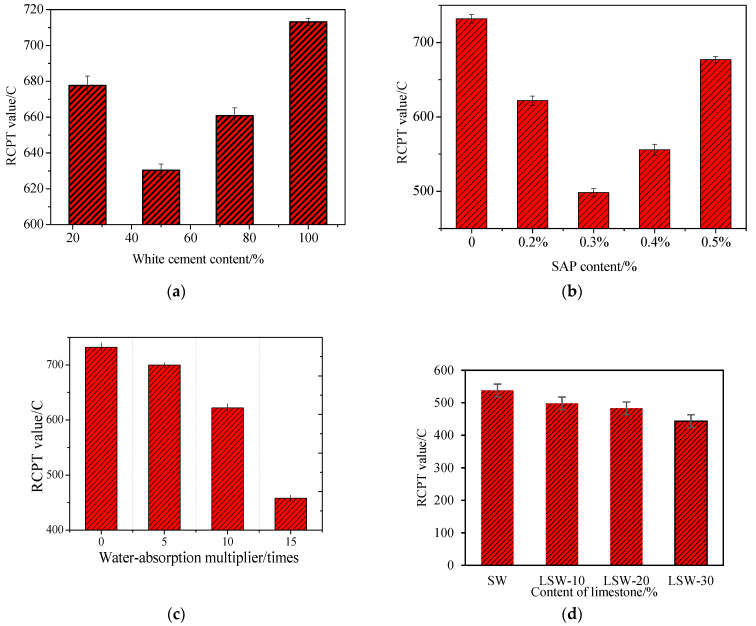
RCPT value of fair-faced concrete; (**a**) RCPT value with different white cement content; (**b**) RCPT value with different SAP content; (**c**) RCPT value with a different water absorption rate; (**d**) RCPT value with a different content of limestone.

**Figure 4 materials-16-07058-f004:**
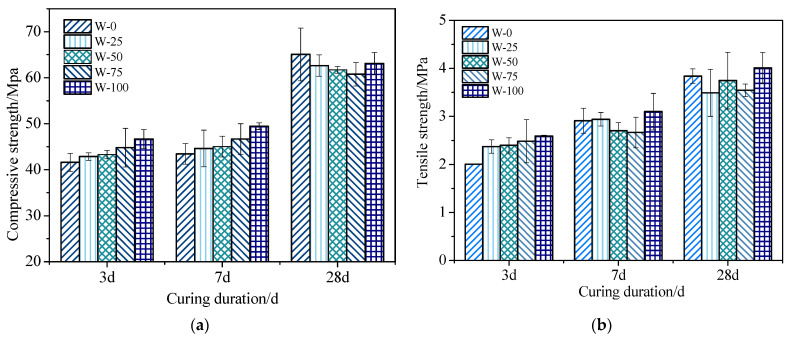
Strength of white cement with fair-faced concrete: (**a**) Compressive strength of white cement with fair-faced concrete; (**b**) tensile strength of white cement with fair-faced concrete.

**Figure 5 materials-16-07058-f005:**
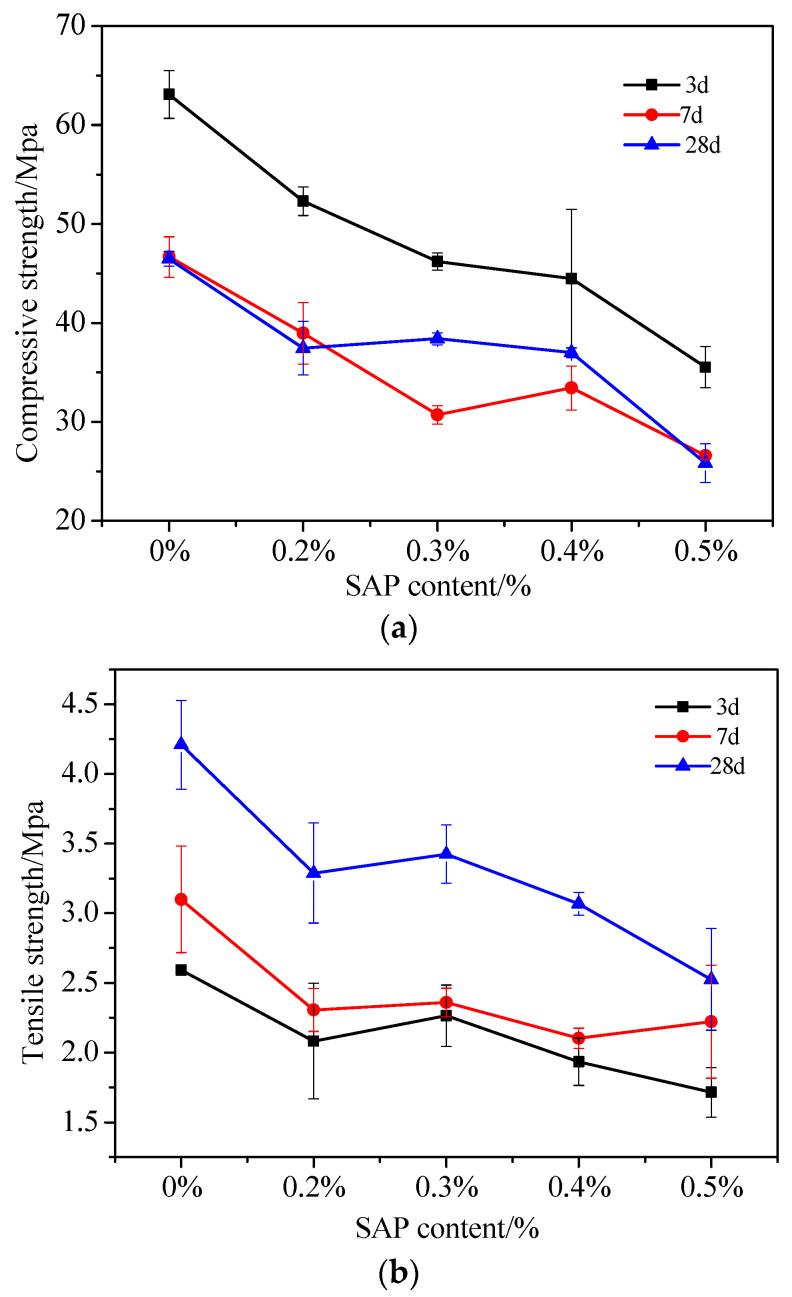
Influence of SAP content on the strength of fair-faced concrete: (**a**) Influence of SAP content on compressive strength; (**b**) influence of SAP content on tensile strength.

**Figure 6 materials-16-07058-f006:**
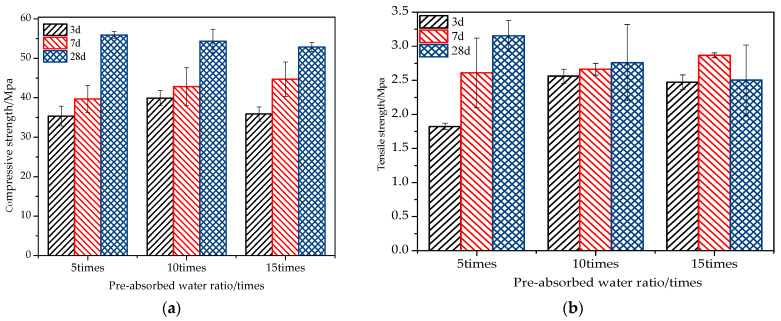
Influence of SAP pre-absorbed water ratio on the strength of fair-faced concrete: (**a**) compressive strength; (**b**) tensile strength.

**Figure 7 materials-16-07058-f007:**
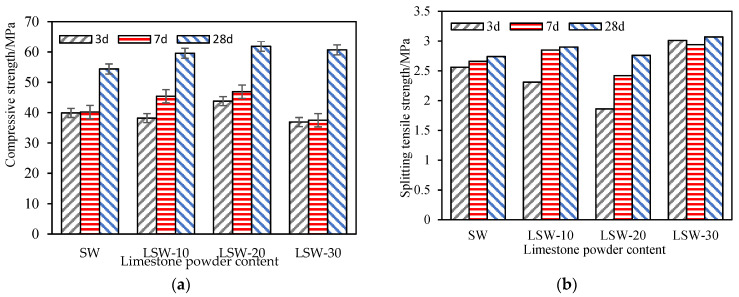
Strength curves of different types of fair-faced concrete: (**a**) Compressive strengths; (**b**) tensile strengths.

**Figure 8 materials-16-07058-f008:**
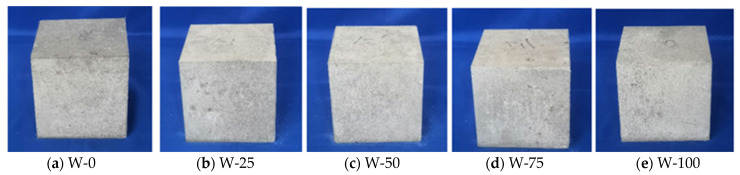
Appearance of fair-faced concrete with different white cement contents. (**a**) W-0: 0% white cement, (**b**) W-25: 25% white cement, (**c**) W-50: 50% white cement, (**d**) W-75: 75% white cement, (**e**) W-100: 100% white cement.

**Figure 9 materials-16-07058-f009:**
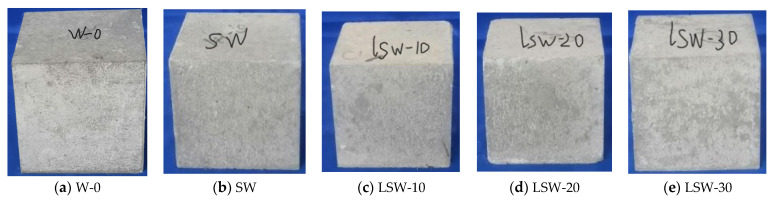
Appearance of fair-faced concrete with different limestone powder contents. (**a**) W-0: 0% white cement, (**b**) SW: 50% white cement, 0.2% SAP, (**c**) LSW-10: 50% white cement, 0.2% SAP, 10% limestone, (**d**) LSW-20: 50% white cement, 0.2% SAP, 20% limestone, (**e**) LSW-30: 50% white cement, 0.2% SAP, 30% limestone.

**Figure 10 materials-16-07058-f010:**
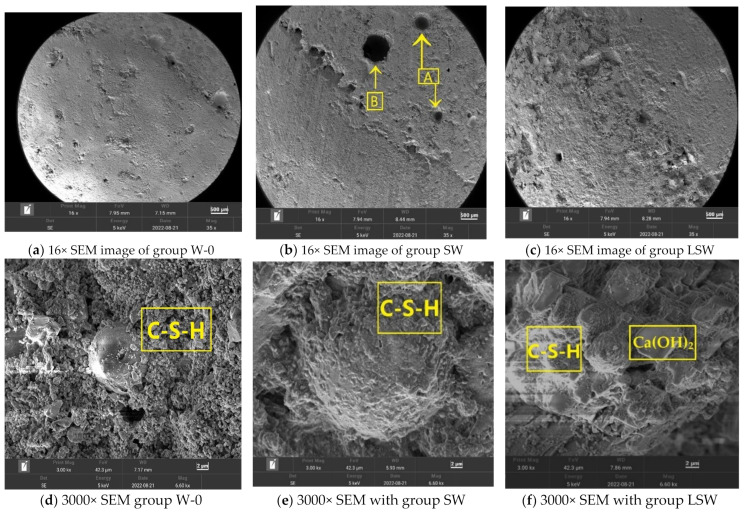
(**a**) 16× SEM image of group W-0; (**b**) 16× SEM image of group SW; (**c**) 16× SEM image of group LSW; (**d**) 3000× SEM image of group W-0; (**e**) 3000× SEM image of group SW; (**f**) 3000× SEM image of group LSW.

**Table 1 materials-16-07058-t001:** Chemical composition of cementitious materials /%.

Component	CaO	SiO_2_	Al_2_O_3_	MgO	Fe_2_O_3_	K_2_O	LOI
White Portland Cement	62.79	20.75	3.22	2.53	0.25	1.34	1.96
Ordinary Portland Cement	52.84	14.24	5.41	1.80	2.46	0.89	9.87
Fly ash	3.32	39.80	28.99	0.52	4.28	1.31	21.28

**Table 2 materials-16-07058-t002:** SAP performance indicators.

Materials	Particle Size (mm)	Deionized Water Absorption (g/g)	Pre-Absorbed Water Ratio (s)	PH	Volume Density (g/cm^3^)
SAP	0.25–0.6	≥200	≤60 s	6~7	0.6~0.9

**Table 3 materials-16-07058-t003:** The mix ratios of fair-faced concrete (Kg/m^3^).

Label	Ordinary Cement	White Cement	Limestone Powder	SAP	Fly Ash	Pre-Absorbed Water	Gravel	Sand	Water	Water Reducer
W-0	317	0	0	0	133	0	1083	752	150	4.6
W-25	237.75	79.25	0	0	133	0	1083	752	150	4.6
W-50	158.5	158.5	0	0	133	0	1083	752	150	4.6
W-75	79.25	237.75	0	0	133	0	1083	752	150	4.6
W-100	0	317	0	0	133	0	1083	752	150	4.6
SW-0.2	158.5	158.5	0	0.9	133	9	1083	752	150	4.6
SW-0.3	0	317	0	1.35	133	13.5	1083	752	150	4.6
SW-0.4	0	317	0	1.8	133	18	1083	752	150	4.6
SW-0.5	0	317	0	2.25	133	22.5	1083	752	150	4.6
SW-5	158.5	158.5	0	0.9	133	9	1083	752	150	4.6
SW-10	158.5	158.5	0	0.9	133	9	1083	752	150	4.6
SW-15	158.5	158.5	0	0.9	133	9	1083	752	150	4.6
LSW-10	136	136	45	0.9	133	9	1083	752	150	4.6
LSW-20	113.5	113.5	90	0.9	133	9	1083	752	150	4.6
LSW-30	91	91	135	0.9	133	9	1083	752	150	4.6

**Table 4 materials-16-07058-t004:** Classification of Concrete Resistance to Chloride Ion Penetration (Electric Flux Method) [53].

Level	I	II	III	IV	V
Electric Flux (Q_s_)/C	Q_s_ ≥ 4000	2000 ≤ Q_s_ ˂ 4000	1000 ≤ Q_s_ ˂ 2000	500 ≤ Q_s_ ˂ 1000	Q_s_ ˂ 500

**Table 5 materials-16-07058-t005:** The *p*-value between each response variable and each factor.

P	Compressive Strength/MPa			Tensile Strength/MPa			RCPT Value/c	Slump/mm
	3 d	7 d	28 d	3 d	7 d	28 d		
White Cement Content	0.6971	0.8212	0.0925	0.3494	0.1692	0.0633	0.4096	0.7844
SAP Content	0.0388	0.0064	0.0000	0.0305	0.1330	0.0093	0.1348	0.0036
Pre-Absorbed Water Ratio	0.2364	0.5315	0.8232	0.2010	0.6324	0.0590	0.0001	0.2276
Limestone Content	0.1875	0.6372	0.0003	0.1632	0.4195	0.7516	0.0537	0.0031

## Data Availability

Data available on request.

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
