# Peer review of "Synergistic Effects of SAP, Limestone Powder and White Cement on the Aesthetic and Mechanical Properties of Fair-Faced Concrete"

_materials, 2023, doi:10.3390/ma16217058_

Round 1
Reviewer 1 Report
Comments and Suggestions for Authors
An interesting article of an applied nature on veneer concrete. Typical research methods were used but the form of conclusions of the obtained results is boring. Basically the same repeated conclusions occur in every subsection. I hope that future articles will be more economical in this regard. With the caveat that perhaps for engineers and designers this is an appropriate form of communicating results. Nevertheless, I congratulate the authors on an interesting article.
Reviewer 2 Report
Comments and Suggestions for Authors
The paper “Synergistic Effects of SAP, Limestone Powder and White Cement on the Aesthetic and Mechanical Properties of Fair-faced Concrete” presents good investigation results, it is well organized and easy to read and deserves a positive overall evaluation.
The English language used in the manuscript is good enough without critical mistakes of grammar, selection of words and typing. Please, check the formatting of the article as it does not match the journal TEMPLATE. Moreover, lines are not numbered, which makes it very difficult to review.
As for the introduction, although it includes a good exposition of the background, there is a clear bias in the citations from the same geographic region. There are multiple researchers from other countries who have conducted related research, please review and cite.
I would recommend eliminating the “P” in the naming of the mixtures, as it is confusing. For example, mixture SWP -0.4 would be SW 0.4 and LSWP-20 would be LSW-20. Reference mixture could be named as desired.
Figure 1 is completely blurry and cannot be read. Some other figures (8 and 9) are blurry and should be enhanced, although that may be a result of the submission process and can be solved in the printing phase.
2.2.2 Strength Test should be “Tests”
The experimental methods should incorporate the methodology of the workability (slump).
It would be valuable to incorporate an ANOVA study to analyze the significance of each effect (section 3.1), as in https://doi.org/10.3390/math8122190.
Minor typos all over the document, please, revise. Check section 3.1.2. What are LSAC-10, LSAC-20, and LSAC-30 ?? Do you mean LSWP? Check legend of figure 6. Check figure caption of figure 7, naming mistakes.
The number of SAMPLES of each test should be explicit.
Conclusions are in accordance with the results, but I would recommend expanding them, to better summarize all the main findings.
Comments on the Quality of English LanguageMinor typos all over the document, please, revise.
Reviewer 3 Report
Comments and Suggestions for Authors
The manuscript needs major revision and before can be recommended for publication. The following comments should be considered in revising the manuscript.
· Please briefly explain how you have chosen SAP content of 0.2% to 0.5%? Why not more than these?
· Give some info on super absorbent resin such as density, specific gravity, etc.
· What was the applied voltage for the RCPT test?
· It is suggested to add a photo for different tests/experiments.
· Figure 1 is not visible. Please improve the quality.
· Please mention what is acceptable RCPT value for concrete to ensure better durability.
· Figure 2b, please discuss the reason for the inconsistency in the results.
· For figure 3a, the reason for lower strength for white cement concrete is discussed by the higher C3S content in page 7. How did you observe it? Or is it found in the literature?
· Figure 5, it is not clear what do you mean by water absorption rates of 5, 10, and 15 to a cement mixture?
· SEM images are not visible. Did you use the same magnification for all images? How do you say LSWP is the best among all? Could you please mark the hydration products on the images to distinguish the images?
Comments on the Quality of English LanguageIt would be better to use english editing facilities to improve the grammatical errors in some places.
Round 2
Reviewer 3 Report
Comments and Suggestions for Authors
Just improve the quality of Figure 1. I have no more comments.